# tRNA-Derived Fragments as Biomarkers in Bladder Cancer

**DOI:** 10.3390/cancers16081588

**Published:** 2024-04-20

**Authors:** Olaf Strømme, Kathleen A. Heck, Gaute Brede, Håvard T. Lindholm, Marit Otterlei, Carl-Jørgen Arum

**Affiliations:** 1Department of Clinical and Molecular Medicine, Norwegian University of Science and Technology, 7034 Trondheim, Norway; kathleen.heck@ntnu.no (K.A.H.); gaute.brede@ntnu.no (G.B.); haavard.lindholm@uhn.ca (H.T.L.); marit.otterlei@ntnu.no (M.O.); carl-jorgen.arum@ntnu.no (C.-J.A.); 2Princess Margaret Cancer Centre, University Health Network, Toronto, ON M5G 1L7, Canada; 3Department of Urology, St. Olav’s University Hospital, 7030 Trondheim, Norway

**Keywords:** tRNA-derived fragment, tRF, EV, exosomes, bladder cancer, biomarker, tRNA

## Abstract

**Simple Summary:**

Diagnosis of bladder cancer is reliant on cystoscopy, which is an invasive procedure. The aim of this study was to investigate the potential of tRNA-derived fragments from noninvasive liquid biopsies as biomarkers in bladder cancer. We identified several tRNA-derived fragments in extracellular vesicles from urine and serum as well as in serum supernatant, which potentially can be used to diagnose disease stages in bladder cancer.

**Abstract:**

Bladder cancer (BC) diagnosis is reliant on cystoscopy, an invasive procedure associated with urinary tract infections. This has sparked interest in identifying noninvasive biomarkers in body fluids such as blood and urine. A source of biomarkers in these biofluids are extracellular vesicles (EVs), nanosized vesicles that contain a wide array of molecular cargo, including small noncoding RNA such as transfer RNA-derived fragments (tRF) and microRNA. Here, we performed small-RNA next-generation sequencing from EVs from urine and serum, as well as from serum supernatant. RNA was extracted from 15 non-cancer patients (NCPs) with benign findings in cystoscopy and 41 patients with non-muscle invasive BC. Urine and serum were collected before transurethral resection of bladder tumors (TUR-b) and at routine post-surgery check-ups. We compared levels of tRFs in pre-surgery samples to samples from NCPs and post-surgery check-ups. To further verify our findings, samples from 10 patients with stage T1 disease were resequenced. When comparing tRF expression in urine EVs between T1 stage BC patients and NCPs, 14 differentially expressed tRFs (DEtRFs) were identified. In serum supernatant, six DEtRFs were identified among stage T1 patients when comparing pre-surgery to post-surgery samples and four DEtRFs were found when comparing pre-surgery samples to NCPs. By performing a blast search, we found that sequences of DEtRFs aligned with genomic sequences pertaining to processes relevant to cancer development, such as enhancers, regulatory elements and CpG islands. Our findings display a number of tRFs that may hold potential as biomarkers for the diagnosis and recurrence-free survival of BC.

## 1. Introduction

Bladder cancer (BC) is one of the most prevalent cancers globally and is associated with substantial morbidity, mortality and costs. Currently, cystoscopy is the primary tool for diagnosing and monitoring BC patients. As cystoscopy is an invasive procedure associated with urinary tract infections [1], there is a significant interest in identifying biomarkers from noninvasive techniques that could aid in BC diagnosis and treatment.

Small noncoding RNA (sncRNa), including t-RNA-derived fragments (tRFs) and microRNA (miRNA), have over the last decade received significant attention as potential cancer biomarkers [2,3]. While transfer RNA (tRNA) was discovered in the late 1950s [4] and fractions of tRNAs were identified in cancer patient urine and serum in the 1970s [5,6], it was not until the deep sequencing era that tRFs were recognized as a distinct sncRNA subclass [7].

The maturation of transfer RNA (tRNA) is a complex process, entailing extensive modifications and processing steps [8]. Initially, RNase Z (ELAC 1/2) and RNase P facilitate the removal of leader sequences at both the 3’ and 5’ ends. Subsequent modifications include TSEN-complex-mediated excision of introns and the addition of a non-templated CCA sequence, which signifies the completion of tRNA maturation [9]. The cleavage of both pre-tRNA and mature tRNA molecules that gives transfer RNA fragments (tRFs) is specific, leading to the presence of the same tRFs. The classification of tRFs into two major categories is determined by their location on the mature tRNA and their origin, whether from the 5’ or 3’ end of the parent tRNA molecule. Additionally, tRNA halves, which are 31–40 nucleotides in length, are produced via cleavage by angiogenin within the anticodon loops of mature tRNAs. Concurrently, tRFs, typically ranging from 14 to 30 nucleotides, originate from cleavages at various sites on either precursor or mature tRNAs. This results in distinct tRF subtypes, including tRNA halves, i-tRF, 3-tRF and 5-tRF (Figure 1A) [8,9,10]. 

One of the main functions of tRFs is the regulation of gene expression. This occurs at the transcriptional level by binding transcription factors [11] and at the post-transcriptional level through RNA interference. Recent data have shown that tRFs are abnormally expressed in multiple malignant tumors and related to tumorigenesis [12]. For instance, tRFs may repress protein translation by binding a complementary mRNA target, resembling miRNAs [13]. Thus, tRFs can potentially function as tumor suppressors and oncogenes, depending on the individual tRF’s nucleotide sequence and complementary target [14,15]. Relatively little is known about the role of tRF in BC pathogenesis, though Su et al. identified a specific tRF methylation pattern in BC cells that attenuated gene silencing of potential proto-oncogenes. tRFs with specific methylation patterns may thus be drivers of oncogenesis in BC [16,17,18].

tRFs may reside in extracellular vesicles (Evs), which are nanosized vesicles secreted from most cell types. Evs can be found in bodily fluids such as blood and urine [8,19]. The term EV refers to two distinct subgroups of vesicles: exosomes, which are small Evs (30–100 nm in diameter) and stem from intraluminal vesicles of the endocytic pathway [20], and microvesicles (MV), stemming from budding of the plasma membrane and which are 100–1000 nm in diameter [21]. Both types of Evs contain molecular cargo, such as mRNA, sncRNA and proteins, which may be transferred from one cell to another. Through the exchange of intravesicular cargo, Evs may play a part in the pathogenesis of a wide range of diseases, including cancer [22,23]. As the rate of EV secretion is higher in cancer cells than healthy cells [24] and the EV cargo is shielded from degradation in bodily fluids such as blood, urine and saliva, there has been significant interest in Evs as reservoirs of cancer biomarkers [25].

Only recently have EV-contained tRFs (EvtRFs) been explored as potential cancer biomarkers. Zhu et al. found a significantly higher level of plasma exosomes in liver cancer patients than in healthy controls and they also identified four exosomal tRFs that could have potential as diagnostic biomarkers in liver cancer patients [26]. Lin et al. identified three exosomal tRFs in plasma of gastric cancer patient plasma, tRF-18, tRF-25 and tRF-38 that were significantly upregulated compared to healthy controls [27]. In breast cancer, EV-contained tRFs have been identified as having potential as diagnostic biomarkers [28,29]. While tRFs have been identified in BC patient urinary Evs [30], a role for EV-contained tRFs in BC diagnosis is largely unexplored.

We previously reported a study identifying potential BC miRNA biomarkers through next-generation sequencing (NGS) of urine Evs, serum Evs and serum supernatant samples from 41 non-muscle invasive BC (NMIBC) patients and 15 noncancer patients (NCPs). In this study, the same patient material and methodology were applied to explore tRFs as potential biomarkers in BC.

## 2. Materials and Methods

### 2.1. Clinical Samples

Samples of serum supernatant and urine were chosen from 15 NCPs and 41 NMIBC patients recruited to the VESCAN biobank project. The biobank comprises urine and serum samples collected for patients who undergo (1) cystoscopy examinations, (2) BC surgery and (3) BC post-surgery check-ups. Characteristic of included NIMBC samples were (1) biobanking of urine and serum occurred prior to BC surgery; (2) no evidence of BC recurrence was detected in at least one post-surgery check-up where urine and serum was biobanked; (3) there was no evidence of concurrent or prior malignancy other than BC. For most patients, samples from an early (normally 3 months) and late post-surgery check-up (normally 12 months) were included. Samples from NCPs were included if patients had no history of malignancy and presented with benign findings on cystoscopy examination. Clinical details for all patients and corresponding samples are described in our previous study [29]. 

### 2.2. Workflow and Study Design

A summary of workflow and study design is outlined in Figure 1B,C. Urine and serum was centrifuged at 500× *g* for 15 min and 12,000× *g* for 30 min and the supernatant stored at −80 degrees before isolation of EV/RNA and subsequent small RNA sequencing. A total of 18 stage Ta and 10 stage T1 (in total 28 BC patients) and 15 NCPs were sequenced in the first sequencing run and 9 stage Ta and 4 stage T1 (in total 13) additional BC patients as well as replicate samples for the 10 stage T1 patients from the first sequencing run were sequenced in a second sequencing run. MINT map was used for tRF identification, which maps all sequencing reads to the complete tRF space. Deseq2 was employed for the statistical analysis. 

### 2.3. RNA Isolation

The exoRNeasy kit (Qiagen, Hilden, Germany) was used to isolate EV-contained RNA from urine and serum supernatant. A total of 0.5 mL of serum supernatant and 5 mL of urine was used. Total RNA was isolated from serum supernatant samples using the miRNeasy kit (Qiagen, Hilden, Germany). 

### 2.4. Small RNA Sequencing

Small RNA sequencing was performed as previously described [31].

### 2.5. Statistical Analysis

Atropos was used to trim adapters with the adapter sequence “TGGAATTCTCGGGTGCCAAGG”. Counts for tRFs were assessed using MINTmap [32]. Differential expression analysis was performed with DESeq2 using the exclusive counts from MINTmap. The parameter biosource_cancer-stage [33] (for example, SERUM_TA) was used for the differential expression, which was performed separately for initial sequencing samples and for replicated samples. Differentially expressed tRFs (DEtRFs) were defined when an adjusted *p*-value was lower than 0.05 between the compared sample groups. The R package eulerr was used to create Venn diagrams and pheatmap was used to create heatmaps. Data from MINTbase were used to assign information about which tRNA each tRF originated from and what type each tRF was. PCA was performed using the PCA function from sklearn. 

### 2.6. BLAST Search

To examine what regions in the genome tRFs could interact with, we performed a BLAST search for each tRF sequence for tRFs of interest. The BLAST search was performed using the nucleotide collection for homo sapiens and the program blastn. Hits longer than 1 million and whole chromosome hits were filtered out as well as hits with “match_diff” <2 and “hsp_gaps” < 2. A list of phrases of interest was created based upon the filtered blast results and the number of hits with these phrases was summarized.

## 3. Results

### 3.1. Sample Characterization

Patient cohort attributes are outlined in detail as previously described [31]. Sample source and processing are outlined in Figure 1B,C. Principal component analysis (PCA) of tRF counts was conducted on serum EVs, urine EVs and serum supernatant for both the initial and replica sequencing runs (Appendix A). Nanoparticle tracking analysis had previously confirmed the isolation of particles consistent with the size of EVs [31]. We noted clustering of the three biosources, though there was some overlap for serum EV and serum supernatant samples. An explanation for the larger separation by urine EV samples may be the higher amount of tRF reads in urine EV samples (Appendix A). A larger fraction of tRFs in urine compared to other biofluids has previously been reported [34].

### 3.2. Differential Expression of tRFs in Urine EVs, Serum EVs and Serum Supernatant in the Initial Sequencing Run

Differentially expressed tRFs (DEtRFs) were identified in both serum EVs, urine EVs and serum supernatant (Figure 2). For urine EVs from Ta patients, 96 DEtRFs were mutually expressed when comparing pre-surgery samples to post-surgery samples, while 48 DEtRFs were found when comparing pre-surgery samples to NCPs. For T1 patients, the opposite trend was observed as 98 DEtRFs were found in the comparison of pre-surgery samples to NCPs and only a single DEtRF was found when comparing pre-surgery to post-surgery samples (Figure 2A). 

In serum EVs from Ta patients, 36 DEtRFs were found in the comparison of pre-surgery samples to post-surgery samples and 31 when comparing pre-surgery to NCP samples. There was a significant overlap of DEtRFs between the two comparisons, as 18 tRFs were mutually differentially expressed. A similar result was observed for T1 patients, where 49 DEtRFs were found in the comparison of pre-surgery to post-surgery samples and 41 when comparing pre-surgery to NCPs, while 32 of these DEtRFs were identified in both comparisons (Figure 2B). In serum supernatant, a total of 32 DEtRFs were identified, of which 13 individual tRFs were mutually differentially expressed in both the pre-surgery vs. post-surgery and pre-surgery vs. NCP comparison (Figure 2C).

To summarize, the number of DEtRFs depends both on the sample source and disease.

### 3.3. tRFs Are Confirmed as Differentially Expressed among Patients with Stage T1 Disease by Replica Sequencing

A replica EV/sncRNA isolation and sequencing run was carried out on new aliquots of urine and serum supernatant from 10 patients with stage T1 disease from the original cohort. Log2 fold change was compared between the initial and replica sequencing for all identified tRFs. A notable correlation was seen in urinary EVs between the two runs with a calculated R-squared value of 0.19 for T1 pre-surgery vs. NCP (Figure 3A) and 0.14 for T1 pre-surgery vs. post-surgery (Figure 3B). In serum supernatant, R-squared values of 0.21 for T1 pre-surgery vs. NCP (Figure 3C) and 0.045 for the T1 pre-surgery vs. post-surgery comparisons were obtained (Figure 3D).

tRFs that were differentially expressed in both sequencing runs were plotted on heatmaps based on log2 fold change and adjusted *p*-value (Figure 4A,B). In urine EVs, 14 tRFs were identified as differentially expressed in both the initial and replica sequencing, all of which were found when comparing T1 pre-surgery samples to NCPs (Figure 4A). In serum supernatant, six DEtRFs were identified in both sequencing runs. Interestingly, four of these were found to be differentially expressed when comparing pre-surgery samples to both post-surgery and NCP samples. Meanwhile, the remaining two samples were differentially expressed solely when comparing pre-surgery samples to NCPs (Figure 4B). 

The 14 tRF sequences for urine EVs and 6 tRF sequences for serum supernatant were aligned using clustal omega. The alignment of the tRFs showed that there are overlapping sequences between the DEtRFs fragments in both biological sources (Figure 4C,D). As MINTmap requires that reads match a tRF exactly in order to be counted, even a single nucleotide difference will change the count associated with a tRF. This would also explain why some tRFs, although similar in sequence, have different log foldchange. However, the sequence similarity between tRFs should be considered when interpreting these results.

### 3.4. tRFs Differentially Expressed between Bladder Cancer and Noncancer Patients Are of Diverse Types and Originate from a Multitude of tRNAs

Each tRF can be categorized depending on where in the tRNA it originates from and which tRNA it is derived from. Here, we used the definitions of tRF type as defined by MINTmap (Figure 1A). Regarding the comparison between pre-surgery and NCP samples, we found DEtRFs across all tRF types except 3’-halves (Figure 5A). However, the distribution is not even and 5’-halves are underrepresented. To investigate whether there was a bias towards a certain origin of the tRFs, we plotted a heatmap showing which parent tRNA each DEtRF originated from (Figure 5B). This heatmap shows that DEtRFs originate from most tRNAs. However, there is an enrichment of DEtRFs originating from certain parental tRNAs, such as tRNAsGlyGCC, which are particularly prevalent. Together, these results show that DEtRFs are of diverse types and originate from a multitude of tRNAs. 

### 3.5. Assessment of Potential Role for tRFs in Biological Processes

tRFs can interact with a range of cellular processes, often mediated by binding through their nucleotide sequence [35]. Furthermore, they have also been found associated with tamoxifen resistance in breast cancer [36]. To generate a hypothesis about which biological processes DEtRFs may be involved in, we performed a blast search using the tRF nucleotide sequences (Appendix A). We chose to focus on DEtRFs identified in patients with stage T1, as these were confirmed as differentially expressed by replica sequencing. A list of interesting phrases obtained by the blast search was manually created for urine EVs (Table 1) and serum supernatant (Table 2). We found the phrase “tRNA” for 12 out of 14 DEtRFs in urine EVs and five out of six DEtRFs in supernatant, confirming the identification of tRFs. 

However, we also obtained several other hits. Notably, three DEtRFs match with piRNA, which can influence cellular function through epigenetic or post-transcriptional silencing. Furthermore, some of the tRFs have matching sequences to enhancer sequences, regulatory elements or CpG islands, which is another indication that they could influence gene regulation. Finally, a number of gene names also turn up in the search such as HES7, which is an important part of the Notch signalling pathway. Further evidence is needed to see if these tRFs can influence such biological processes and to confirm that the sequences we find in our data originate from tRFs and not these non-tRF sequences themselves.

## 4. Discussion

tRF is a recently acknowledged class of sncRNA that have been shown to play a role in carcinogenesis and may have potential as cancer biomarkers. In the present study, the main findings can be summarized in two main points: (1) in both urine EVs and serum supernatant, we identified DEtRFs that were confirmed through replica sequencing; (2) sequences of DEtRFs were found to be aligned to genomic sequences linked to processes associated with carcinogenesis, such as enhancers, regulatory elements and CpG islands.

Taking replica sequencing results into account, we identified 14 DEtRFs in urine EVs. Of these, 12 were upregulated and 2 downregulated when pre-surgery samples were compared to NCP samples. Three of the tRFs that were upregulated, tRF-18-MBQ4NKDJ, tRF-20-40KK5Y93 and tRF-31-PER8YP9LON4VD, have previously been identified in cancer biomarker studies. In a study of gastric cancer patients, the expression of both tRF-18-MBQ4NKDJ and tRF-31-PER8YP9LON4VD was significantly upregulated in gastric cancer patient plasma compared to healthy controls [35]. In a study on tRFs in breast cancer, however, the expression of tRF-20-40KK5Y93 and tRF-31-PER8YP9LON4VD was downregulated in breast cancer tissue, as compared to healthy adjacent tissue [37]. Regarding results from serum supernatant, six DEtRFs were identified when comparing pre-surgery samples to NCPs and four DEtRFs when comparing pre-surgery to post-surgery samples. 

In similar ways to other sncRNAs, tRFs can regulate cellular processes through complementary sequence binding in processes such as RNA silencing. In one interesting example from bladder cancer, m1 A-modified tRF-3004b was shown to interact with the unfolded protein response [38]. In this way, tRFs can potentially influence cancer progression depending on their sequence. In support of these DEtRFs influencing cancer progression, the BLAST search performed here revealed overlap with cellular processes regulating gene regulation and genes involved in the development of cancer. These hits include enhancer regions, CpG islands and regulatory elements which are nonmutational epigenetic processes and, in some cases, can be an enabling characteristic in oncogenesis [39]. The association with enhancer regions may be of particular importance. Interestingly, mammalian-wide interspersed tandem repeats (MIR), which are conserved parts of the human genome that are derived from tRNAs, have been found to act as enhancers of gene expression. It would thus be of particular interest if DEtRFs found in our study could directly interact with MIRs, potentially affecting the regulation of a wide set of genes [40]. Other hits are found in genes such as FCGR2A, GPD2 and HES7. Some care should be taken when interpreting the number of hits in the BLAST search results as some of the DEtRFs have sequence overlap, as shown in Figure 4C,D. Unlike MINTmap, the BLAST algorithm allows for some mismatches in the nucleotide sequence and, therefore, two similar tRFs could overlap with the same hits in the BLAST search results. 

As cancer normally develops after changes in more than one gene, it can be difficult to establish the exact role of changes in these genes in the development of cancer. However, the gene HES7 is part of the notch signaling pathway, which can influence bladder cancer progression [41]. In support of tRFs affecting the notch signaling pathway, a tRF named CAT1 has been found to regulate the stability of NOTCH2 mRNA and promote tumorigenesis [42]. It is interesting that 6 out of 20 DEtRFs overlap with sequences annotated as piRNAs. piRNAs are the same length as tRFs and some of the piRNAs are complete matches to the tRFs. piRNAs can guide proteins to cleave target RNA, methylate DNA and promote heterochromatin assembly and it would therefore be interesting to investigate how tRFs might influence their biology [43]. However, a hit in the BLAST search does not necessarily mean that there is a functional connection between the DEtRF and the annotated biological process. It is possible that the tRF counting algorithm match reads in the sequencing data to tRFs that do not originate from tRFs or that the annotation in the database used for the blast search is wrong. The results from the BLAST search are therefore best used to generate hypotheses and future functional studies should therefore investigate the possible connection between the DEtRFs discussed here and cancer progression. 

While previous studies have investigated tRF expression in BC tissue among muscle-invasive bladder cancer (MIBC) patients [44] and tRFs in early disease progression vs. poor treatment outcome in BC [45], the present work is, to the best of our knowledge, the first to identify EV-contained tRFs as potential biomarkers in BC. Some of the study’s limitations need to be addressed, however. The patient sample size is small, with the stage T1 group consisting of 14 patients and the NCP group consisting of 15 patients. Furthermore, only samples from T1 patients were subjected to a replica EV/RNA isolation and sequencing run, limiting the basis for identifying biomarkers in the Ta group. 

Here, 14 DEtRFs in urine EVs and 6 DEtRFs in serum supernatant were identified as potential diagnostic biomarkers for BC patients with stage T1 disease. Four of the DEtRFs identified in serum supernatant, tRF-16-F1R3WEE, tRF-18-F8DHXYD9, tRF-19-DRXSE5I2 and tRF-29-KY7SHRRNWJE2, were downregulated in pre-surgery samples, both compared to post-surgery and NCP samples, suggesting this downregulation may be specific to BC patients with stage T1 disease. As post-surgery samples were collected from patients with no clinically detected recurrence at the time of sampling, these four tRFs may hold potential as biomarkers of both primary diagnosis and recurrence-free survival of T1 patients.

## 5. Conclusions

In the present study, we identified 14 DEtRFs urine EVs and 6 DEtRFs serum supernatants from BC patients with stage T1 disease. A number of these DEtRFs were found to be associated with parts of the genome involved in key carcinogenic processes. We therefore argue that the DEtRFs identified in this work may warrant further investigation into their potential as BC biomarkers and as players in BC pathogenesis.

## Figures and Tables

**Figure 1 cancers-16-01588-f001:**
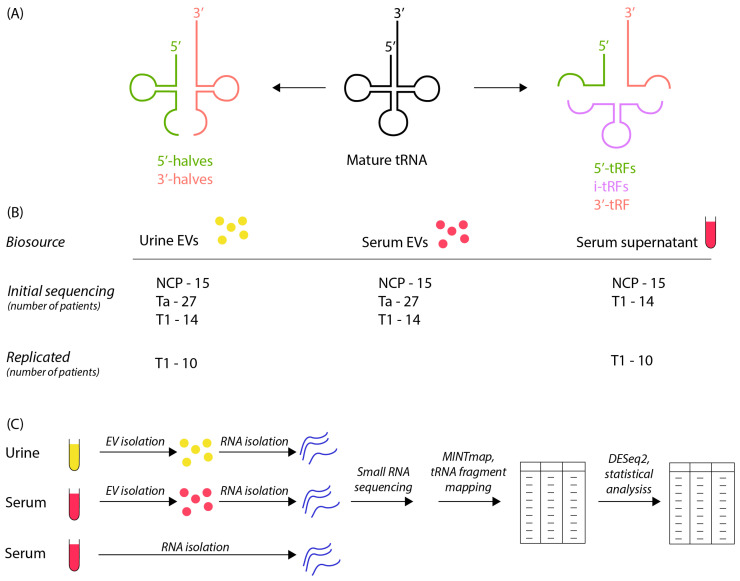
Processing of tRNA-derived fragments, study design and workflow: (**A**) schematic diagram of processing of tRNA-derived fragments from mature tRNA to tRNA halves and tRFs as defined by MINTmap. (**B**) Outline of study design. Urine EVs, serum Evs and serum supernatant samples for noncancer patients (NCP), stage Ta and stage T1 patients were subjected to an initial sequencing run, while urine Evs and serum supernatant samples from 10 T1 patients were included in a replica sequencing run. (**C**) Study workflow, as depicted in the following order: sample collection, EV/RNA isolation, small-RNA sequencing and statistical analysis.

**Figure 2 cancers-16-01588-f002:**
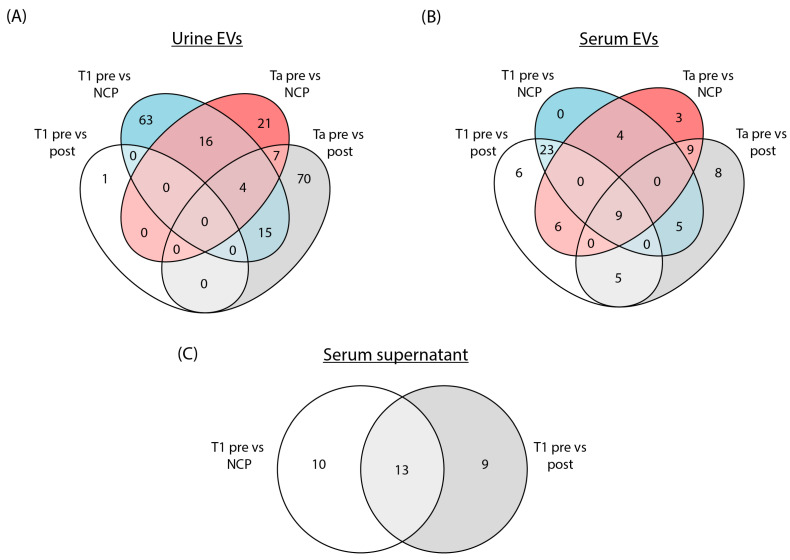
Differential expression of tRNA-derived fragments in urine EVs, serum EVs and serum supernatant. Venn diagrams showing the number of DEtRFs in pre-surgery (pre) vs. post-surgery (post) samples and pre-surgery vs. noncancer patient (NCP) samples for (**A**) urine EVs, (**B**) serum EVs and (**C**) serum supernatant. An adjusted *p*-value threshold of 0.05 was set to determine DEtRFss for all analyses.

**Figure 3 cancers-16-01588-f003:**
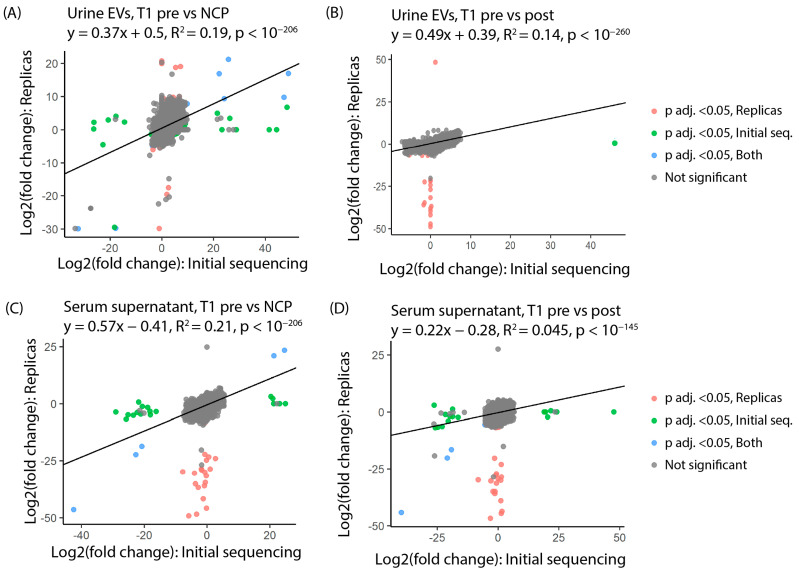
Reproducibility of sequencing data from urine EVs and serum supernatant for T1 patients. The reproducibility of sequencing data from urine EVs (**A**,**B**) and serum supernatant (**C**,**D**) is illustrated by an R-squared value (above the diagram). Additionally, each dot in the diagram represents the log2fold change for an individual tRF in the initial sequencing run (x-axis) and replica sequencing (y-axis), and the dot color indicates whether an individual tRF is differentially expressed in the initial sequencing (green), replica sequencing (orange) or both (blue).

**Figure 4 cancers-16-01588-f004:**
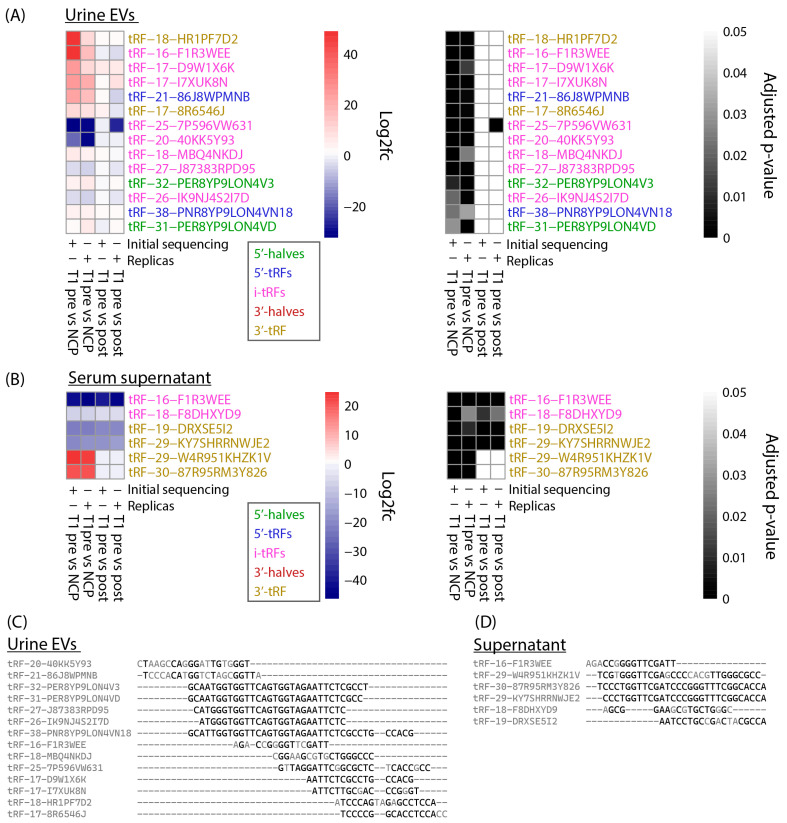
DEtRFs confirmed by replica sequencing. Heatmaps displaying log2foldchange (left) and adjusted *p*-value (right) of DEtRFs confirmed by replica sequencing for (**A**) urine EVs and (**B**) serum supernatant. An adjusted *p*-value threshold of 0.05 was set to determine DEtRFs. Note that plus (+) and minus (−) indicate if the row applies to the initial sequencing or replica sequencing run. (**C**) Overlap of tRF sequences in urine shown in panel (**A**) calculated with clustal omega. (**D**) Overlap of tRF sequences in supernatant shown in panel (**B**) calculated with clustal omega. Nucleotides colored black overlap with a nucleotide in at least one other tRF.

**Figure 5 cancers-16-01588-f005:**
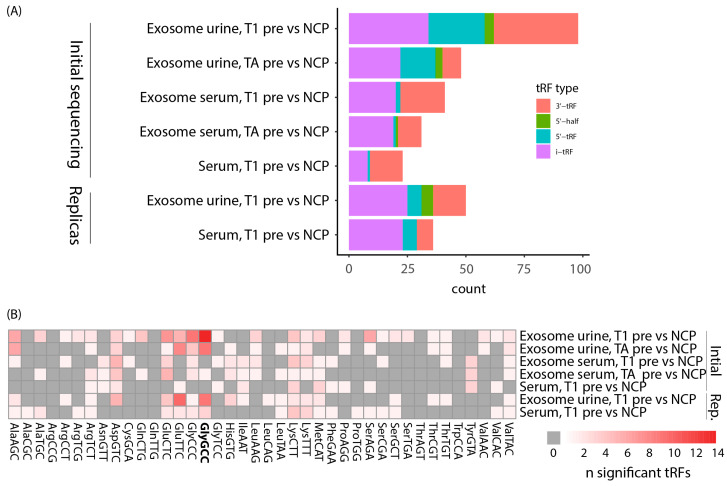
Distribution of DEtRF subtype and parental tRNA source. (**A**) Bar chart showing the distribution of tRF subtype among DEtRFs identified when comparing pre-surgery samples to noncancer patients (NCP) samples. (**B**) Heatmap representation of the parental tRNA source of DEtRFs when comparing pre-surgery and NCP samples. GlyGCC is highlighted as it is discussed in the main text.

**Table 1 cancers-16-01588-t001:** Annotated sequences with matches in DEtRFs from urine EVs. A blast search was performed using the nucleotide sequence for all DEtRFs in Figure 4A. *n* indicates the number of the tRFs with blast results containing the phrase specified here.

Phrase	*n*	tRF Names
FCGR2A	6	tRF-38-PNR8YP9LON4VN18, tRF-32-PER8YP9LON4V3, tRF-31-PER8YP9LON4VD, tRF-27-J87383RPD95, tRF-26-IK9NJ4S2I7D, tRF-17-D9W1X6K
GPD2	6	tRF-38-PNR8YP9LON4VN18, tRF-32-PER8YP9LON4V3, tRF-31-PER8YP9LON4VD, tRF-27-J87383RPD95, tRF-26-IK9NJ4S2I7D, tRF-17-D9W1X6K
H3K4me1 hESC enhancer	11	tRF-38-PNR8YP9LON4VN18, tRF-32-PER8YP9LON4V3, tRF-31-PER8YP9LON4VD, tRF-27-J87383RPD95, tRF-26-IK9NJ4S2I7D, tRF-25-7P596VW631, tRF-20-40KK5Y93, tRF-17-I7XUK8N, tRF-17-D9W1X6K, tRF-17-8R6546J, tRF-16-F1R3WEE
HES7	6	tRF-38-PNR8YP9LON4VN18, tRF-32-PER8YP9LON4V3, tRF-31-PER8YP9LON4VD, tRF-27-J87383RPD95, tRF-26-IK9NJ4S2I7D, tRF-17-D9W1X6K
HSPA6	6	tRF-38-PNR8YP9LON4VN18, tRF-32-PER8YP9LON4V3, tRF-31-PER8YP9LON4VD, tRF-27-J87383RPD95, tRF-26-IK9NJ4S2I7D, tRF-17-D9W1X6K
lncRNA	7	tRF-27-J87383RPD95, tRF-26-IK9NJ4S2I7D, tRF-25-7P596VW631, tRF-21-86J8WPMNB, tRF-18-MBQ4NKDJ, tRF-17-D9W1X6K, tRF-16-F1R3WEE
LTA4	1	tRF-16-F1R3WEE
piRNA	3	tRF-27-J87383RPD95, tRF-26-IK9NJ4S2I7D, tRF-18-MBQ4NKDJ
regulatory element	7	tRF-38-PNR8YP9LON4VN18, tRF-32-PER8YP9LON4V3, tRF-31-PER8YP9LON4VD, tRF-27-J87383RPD95, tRF-26-IK9NJ4S2I7D, tRF-18-MBQ4NKDJ, tRF-17-D9W1X6K
tRNA	12	tRF-38-PNR8YP9LON4VN18, tRF-32-PER8YP9LON4V3, tRF-31-PER8YP9LON4VD, tRF-27-J87383RPD95, tRF-26-IK9NJ4S2I7D, tRF-25-7P596VW631, tRF-21-86J8WPMNB, tRF-20-40KK5Y93, tRF-18-MBQ4NKDJ, tRF-17-I7XUK8N, tRF-17-D9W1X6K, tRF-16-F1R3WEE
VAC14	6	tRF-38-PNR8YP9LON4VN18, tRF-32-PER8YP9LON4V3, tRF-31-PER8YP9LON4VD, tRF-27-J87383RPD95, tRF-26-IK9NJ4S2I7D, tRF-17-D9W1X6K

**Table 2 cancers-16-01588-t002:** Annotated sequences with matches from DEtRFs from serum supernatant. Similar to Table 1, see that description.

Phrase	*n*	tRF Names
H3K4me1 hESC enhancer	2	tRF-29-W4R951KHZK1V, tRF-16-F1R3WEE
lncRNA	3	tRF-29-W4R951KHZK1V, tRF-18-F8DHXYD9, tRF-16-F1R3WEE
LTA4	1	tRF-16-F1R3WEE
piRNA	3	tRF-30-87R95RM3Y826, tRF-29-KY7SHRRNWJE2, tRF-18-F8DHXYD9
regulatory element	1	tRF-18-F8DHXYD9
tRNA	5	tRF-30-87R95RM3Y826, tRF-29-KY7SHRRNWJE2, tRF-19-DRXSE5I2, tRF-18-F8DHXYD9, tRF-16-F1R3WEE

## Data Availability

The data presented in this study are available upon request from the corresponding author.

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
