# Peer review of "tRNA-Derived Fragments as Biomarkers in Bladder Cancer"

_cancers, 2024, doi:10.3390/cancers16081588_

Round 1

Reviewer 1 Report

Comments and Suggestions for Authors

My assessment of the article:

- topical and novel topics, search for new biomarkers of bladder cancer,

- use of molecular biology methods (RNA isolation and small RNA sequencing).

Author Response

Dear reviewer,

Thank you for the feedback. We appreciate that you find the article to discuss a topical and novel topic and that the method used is appropriate.

Reviewer 2 Report

Comments and Suggestions for Authors

Olaf et al presented a clinical study to validate tRNA-derived fragments as biomarkers in bladder cancer. Authors conducted evaluation of biomarkers from urine and serum supernatant samples of 41 NMIBC-patients and 15 NCPs recruited to the VESCAN biobank project and it was well-designed. The manuscript is well presented with all required authenticated approvals. Authors identified 14 DEtRFs urine EVs and 6 DEtRFs serum supernatants  from BC patients with stage T1 disease. Authors concluded that further investigations to claim  the potential as BC-biomarkers and role in BC pathogenesis. 

Following minor revision suggested:-

1. Plagiarism detected in the following sections and needs modifications 

(a). Middle part of abstract

(b). Introduction part-paragraph number 4,

(c) sections 2.1, 2.3, 

2. References need to have a few more reports from the last 3 years 

Author Response

Dear Reviewer,

Thank you for reviewing our manuscript and we appreciate that you find the manuscript well presented and the study well-designed.

Following minor revision suggested:-

  1. Plagiarism detected in the following sections and needs modifications 

(a). Middle part of abstract

(b). Introduction part-paragraph number 4,

(c) sections 2.1, 2.3, 

Thank you for bringing up the fact that some parts of the manuscript have similarities to previous research, we apologize for this oversight and have changed the text in question, please see track changes in the manuscript.

  1. References need to have a few more reports from the last 3 years 

We appreciate the feedback and have updated the manuscript with four references from the last 3 years in the following places. New references are highlighted in red.

Section 1.:

“Small non-coding RNA (sncRNa), including t-RNA-derived fragments (tRFs) and microRNA (miRNA), have over the last decade received significant attention as potential cancer biomarkers [2,3].”

Section 3.5:

Furthermore, they have also been found associated with tamoxifen resistance in breast cancer [36].

Section 4:

In one interesting example from bladder cancer, m1 A-modified tRF-3004b was shown to interact with the unfolded protein response [38].”

“In support of tRFs affecting the notch signaling pathway, a tRF named CAT1 has been found to regulate the stability of NOTCH2 mRNA and promote tumoregenisis [42].”

Reviewer 3 Report

Comments and Suggestions for Authors

Dear Authors,

Thank you very much for reviewing your manuscript. I give you the following question to address in your manuscript, which enhances the understanding and readability of the researcher.

1. What are extracellular vesicles (EVs) and how do they contribute to the identification of biomarkers for bladder cancer diagnosis?

2. How was small-RNA next-generation sequencing utilized to analyze urine EVs, serum EVs, and serum supernatant from patients with non-muscle invasive bladder cancer (BC)?

3. What were the key findings regarding the expression levels of transfer RNA-derived fragments (tRFs) in urine EVs, serum EVs, and serum supernatant samples collected from patients with BC compared to non-cancer patients (NCP)?

4. Can you elaborate on the comparison of tRF expression levels between different stages of BC patients (Ta vs. T1) and non-cancer patients?

5. How were the identified differentially expressed tRFs (DEtRFs) validated and what potential implications do they hold as biomarkers for BC diagnosis and recurrence-free survival?

6. What were the outcomes of the blast search conducted to align DEtRF sequences with genomic sequences relevant to cancer development processes?

7. In what ways could the discovery of these tRFs contribute to improving the diagnostic accuracy and management of bladder cancer?

Best Regards

Author Response

Reviewer 3

Dear Reviewer,

Thank you for the feedback and for reviewing our manuscript. We have answered each question below with relevant citations from the manuscript.  

  1. What are extracellular vesicles (EVs) and how do they contribute to the identification of biomarkers for bladder cancer diagnosis?

Thank you for raising this important point. EVs are nanosized vesicles secreted by most cell types which contains cargo from the cell such as RNA molecules and proteins. As EV cargo represents internal structures of cells as well as being protected from degradation in body fluids, EVs represents an interesting source of biomarkers for bladder cancer diagnosis. This is discussed in the quoted paragraph of the introduction:

“tRFs may reside in extracellular vesicles (EVs) which are nanosized vesicles secreted from  most cell types. EVs can be found in bodily fluids such as blood and urine [8,19]. The term EV refers to two distinct subgroups of vesicles; exosomes, which are small EVs (30-100 nm in diameter) and stem from intraluminal vesicles of the endocytic pathway [20], and microvesicles (MV) stemming from budding of the plasma membrane and which are 100-1000 nm in diameter [21]. Both types of EVs contain molecular cargo, such as mRNA, sncRNA and proteins, which may be transferred from one cell to another. Through the exchange of intravesicular cargo, EVs may play a part in the pathogenesis of a wide range of diseases, including cancer [22,23]. As the rate of EV-secretion is higher in cancer cells than healthy cells [24], and the EV-cargo is shielded from degradation in bodily fluids such as blood, urine and saliva, there has been significant interest in EVs as reservoirs of cancer biomarkers [25].”

  1. How was small-RNA next-generation sequencing utilized to analyze urine EVs, serum EVs, and serum supernatant from patients with non-muscle invasive bladder cancer (BC)?

Thank you for the question. Small-RNA next-generation sequencing allows detection of short RNA such as miRNA and tRFs. The complete method is too extensive to quote in its entirety, but we will shortly summarize our approach. Further details can be found in our method section.

Samples of serum and urine were collected from non-muscle invasive bladder cancer patients (see section 2.1), then EVs or supernatant from serum was extracted (see section 2.2). Total RNA was isolated from EVs or supernatant (see section 2.3). small RNA sequencing was performed to measure RNA quantity levels, to minimize duplication of text this is explained in detail in our previous article, see section 4.6 in reference 30. MINTmap was used to count reads aligning to tRFs and the R-package DESeq2 was used for differential expression analysis (see section 2.5).

  1. What were the key findings regarding the expression levels of transfer RNA-derived fragments (tRFs) in urine EVs, serum EVs, and serum supernatant samples collected from patients with BC compared to non-cancer patients (NCP)?

Our key finding is that we can define several tRFs in urine and serum supernatant which hold potential as biomarkers for the diagnosis and recurrence-free survival of BC. This shows both the potential of using tRFs for diagnosis and establishing disease progression of BC and describes specific tRFs which should be further investigated for diagnosing BC patients. The list of specific tRFs can be found in figure 4 and our conclusion outlines our key finding:

“In the present study, we identified 14 DEtRFs urine EVs and 6 DEtRFs serum supernatants from BC patients with stage T1 disease. A number of these DEtRFs were found to be associated with parts of the genome involved in key carcinogenic processes. We therefore argue that the DEtRFs identified in this work may warrant further investigation into their potential as BC-biomarkers and as players in BC pathogenesis.”

  1. Can you elaborate on the comparison of tRF expression levels between different stages of BC patients (Ta vs. T1) and non-cancer patients?

We appreciate the question as there is a large number of possible sample group comparisons. The sample groups we elected to compare are outlined in figure 2. To investigate possible biomarkers which could indicate disease outcome we choose to compare pre-surgery samples to recurrence free post-surgery samples (for example T1 pre vs post). To investigate possible biomarkers of disease we decided to compare pre-surgery samples to non-cancer patients (NCPs) (for example T1 pre vs NCP). We chose not to compare different stages as our focus at this point was to discover biomarkers related to disease progression, not types of bladder cancer. However, this is an interesting research question and should be investigated in the future.

As mentioned when discussing our key finding above, 14 DEtRFs from urine EVs and 6 DEtRFs from serum supernatant where identified when comparing T1 pre-surgery to NCP or post-surgery (see figure 4). We only have replicated sequencing data from T1 patients, and future studies would need to investigate how hits from Ta patients are replicated.

  1. How were the identified differentially expressed tRFs (DEtRFs) validated and what potential implications do they hold as biomarkers for BC diagnosis and recurrence-free survival?

To validate our findings, we re-isolated EVs or Serum from the same 10 T1 patients, extracted RNA and sequenced them in a new sequencing run. The results of the comparison of these two sample runs can be found in figure 3 and 4. The fact that these replicated samples were extracted independently means that any DEtRF identified in both the analysis of the initial sequencing run and the replicated sequencing run are independent of sample extraction and sequencing. This indicates that it is the underlying biology which is responsible for the observed differential expression of these DEtRFs. However, it would be beneficial to validate these findings in a larger sample cohort in the future.

Our hope is that the identified DEtRFs can be used in future cancer diagnostics tests where the measurement of their level informs diagnosis and treatments options.

  1. What were the outcomes of the blast search conducted to align DEtRF sequences with genomic sequences relevant to cancer development processes?

Thank you for the question. The blast search revealed which parts of the genome the identified DEtRFs align with. As there is potential that tRFs aligned with parts of the genome which overlaps with the tRF sequence, this list of features indicates which biological processes these tRFs potentially can interact with in the cell. However, it is important to emphasize that such interactions need to be verified in future work. The outcome of the blast search is described in the following paragraph from the result section:

“Notably, 3 DEtRFs match with piRNA which can influence cellular function through epigenetic or post-transcriptional silencing. Furthermore, some of the tRFs have matching sequences to enhancer sequences, regulatory elements or CpG islands which is another indication that they could influence gene regulation. Finally, a number of gene names also turn up in the search such as HES7 which is an important part of the Notch signalling pathway. Further evidence is needed to see if these tRFs can influence such biological processes and to confirm that the sequences we find in our data originate from tRFs and not these non-tRF sequences themselves.”

  1. In what ways could the discovery of these tRFs contribute to improving the diagnostic accuracy and management of bladder cancer?

Thanks for bringing up this point. This question somewhat overlaps with question 3 about our key findings and please see the quoted conclusion section in response to that question to see how this is discussed in the manuscript.

Our key finding is that we describe tRFs which are significantly associated with disease status in bladder cancer. Our hope is that in the future measuring the level of tRFs in body fluids will be part of standard diagnostic procedures and inform patient care. An attractive aspect of measuring tRF levels in body fluids is that it is non-invasive procedure. However, further research is needed to validate these findings and to get the necessary data to include the measurement of tRFs in diagnosis of bladder cancer.

Best Regards